# Nailing down Clinical Nuances. Comment on Crotti et al. A Terbinafine Sensitive *Trichophyton indotineae* Strain in Italy: The First Clinical Case of *tinea corporis* and onychomycosis. *J. Fungi* 2023, *9*, 865

**DOI:** 10.3390/jof10030231

**Published:** 2024-03-21

**Authors:** Inigo Navarro-Fernandez

**Affiliations:** Department of Dermatology, University Hospital of Réunion, 97400 Saint-Denis, France; fernandez.navarro@chu-reunion.fr; Tel.: +262-693-64-97-48

**Keywords:** onychomycosis, tinea, trichophytosis, *Trichophyton indotineae*

I have read the paper “A Terbinafine Sensitive *Trichophyton indotineae* Strain in Italy: The First Clinical Case of tinea corporis and onychomycosis” by Crotti et al. with great interest [1]. The authors report a new case of *T. indotineae* infection in Italy, which is presented in a well-organized and nicely illustrated manner. Their emphasis on the growing significance of this emerging pathogen as a global health concern is noteworthy.

The authors assert that they are presenting the first case of onychomycosis caused by *T. indotineae*. They indicate that via direct examination, the skin scraping results were positive, yet it remains unclear if the nail samples also tested positive, or if their claim relies solely on the association of nail alterations with the identification of *T. indotineae* in the skin’s eruption and the resolution of nail changes after treatment. In the photos provided, we can only observe onychomadesis on the fourth and fifth fingers of the left hand, and some degree of discoloration of the nail plate on the second toe of the right foot. Although characteristic nail changes sometimes do not develop until months after the initial fungal invasion, it is important to highlight that neither of these findings is particularly suggestive of onychomycosis [2]. Moreover, onychomadesis typically subsides once the underlying cause resolves (the authors acknowledge that they had limited information on the patient’s medical history due to language barriers). Given the absence of prior descriptions of onychomycosis in T. tineae and the reliance on clinical features not typical of onychomycosis, I argue that alternative causes of nail dystrophy are more plausible, and nail infection by *T. indotineae* should not be assumed in this case.

While microscopic direct examination, sample culturing, and molecular techniques constitute the gold standard for the diagnosis of tinea, I firmly believe that the fundamental semiotics of skin fungal infections remain a crucial asset that should not be overlooked.

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
