# Peer review of "Nailing down Clinical Nuances. Comment on Crotti et al. A Terbinafine Sensitive Trichophyton indotineae Strain in Italy: The First Clinical Case of tinea corporis and onychomycosis. J. Fungi 2023, 9, 865"

_jof, 2024, doi:10.3390/jof10030231_

Round 1

Reviewer 1 Report

Comments and Suggestions for Authors

The author raised a very important question: Is it justified to call nail alterations an onychomycosis just because the patient has a tinea corporis? I strongly agree with the writer of this letter that this is not enough for a sound diagnosis. Second, onychomadesis is the result of a complete standstill of nail growth, which is not a feature of onychomycosis. Onychomadesis is - in most cases - the result of an acute event and therefore it grows out by itself independent from what you do or not.

Author Response

Dear Reviewer 1,

Thank you for your insightful feedback and for supporting our viewpoint on the cautious approach required in diagnosing onychomycosis. Your agreement on the importance of distinguishing between different causes of nail distrophy and the necessity for a rigorous diagnostic approach is highly valued.

Reviewer 2 Report

Comments and Suggestions for Authors

The author raises an important point that an accurate diagnosis of onychomycosis requires confirmatory mycology testing as this condition can often mimic non-fungal nail diseases. However, the statement that the diagnosis was ‘founded on the correlation of nail alternatives, the identification of T. indotineae in the skin’s eruption, and the observed resolution of nail changes following treatment’ is a misrepresentation of the work conducted by Crotti and colleagues. Under Case Description, Crotti et al. states that ‘skin specimens and proximal affected nails were sampled for fungal examination’ and ‘the samples were cultured on Dermasel Agar’. The dermatophyte isolate was subsequently subjected to ITS sequencing for the identification of T. indotineae and multiplex PCR for the detection of terbinafine resistance mutations. Due to the slow-progressive nature of onychomycosis, it is entirely possible that the characteristic abnormal nail changes won’t develop until months after the initial fungal invasion. There may have been some confusion since Crotti et al. only showed culture results for the skin specimen and do not indicate if the sequencing/PCR results came from the skin or the nail isolate.

Comments on the Quality of English Language

I find some sentences to be difficult to read; English editing may be needed.

Author Response

We are grateful to Reviewer 2 for their insightful feedback and for emphasizing the critical role of confirmatory mycology testing when evaluating nail alterations.

My statement was based on the fact that the authors specify direct examination was carried out on "skin scraping samples," followed immediately by "The samples were cultured on Dermasel Agar," suggesting that the culture and sequencing/PCR results come from the skin scrapings. However, I concur with the reviewer that the confusion may arise from not specifying the results' origin. I have now clarified this in my text, amending the statement that the reviewer considered a misrepresentation of Crotti et al.'s work (lines 16-19).

The reviewer also notes, "Due to the slow-progressive nature of onychomycosis, it is entirely possible that characteristic abnormal nail changes won’t develop until months after the initial fungal invasion." I agree and have included the statement "Characteristic abnormal nail changes sometimes won’t develop until months after the initial fungal invasion" in the text, acknowledging this perspective (lines 22-23). Nonetheless, I believe it's important to note that the presence of onychomadesis may suggest other, mostly systemic, diseases rather than onychomycosis.

We hope that these revisions and clarifications will address your concerns. We are grateful for the opportunity to enhance our work through this constructive dialogue.